# OBJECTIVE-AGNOSTIC ENHANCEMENT OF MOLECULE PROPERTIES VIA MULTI-STAGE VAE

## ABSTRACT

Variational autoencoder (VAE) is a popular method for drug discovery and various architectures and pipelines have been proposed to improve its performance. However, VAE approaches are known to suffer from poor manifold recovery when the data lie on a low-dimensional manifold embedded in a higher dimensional ambient space (Dai and Wipf, 2019). The consequences of it in drug discovery are somewhat under-explored. In this paper, we explore applying a multi-stage VAE approach, that can improve manifold recovery on a synthetic dataset, to the field of drug discovery. We experimentally evaluate our multi-stage VAE approach using the ChEMBL dataset and demonstrate its ability to improve the property statistics of generated molecules substantially from pre-existing methods without incorporating property predictors into the training pipeline. We further fine-tune our models on two curated and much smaller molecule datasets that target different proteins. Our experiments show an increase in the number of active molecules generated by the multi-stage VAE in comparison to their one-stage equivalence. For each of the two tasks, our baselines include methods that use learned property predictors to incorporate target metrics directly into the training objective and we discuss the complications that arise with this methodology.

## 1 INTRODUCTION

The use of generative models in the domain of drug discovery has recently seen rapid progress. These methods can leverage large-scale molecule archives describing the structure of existing drugs to synthesize novel molecules with similar properties as potential candidates for future drugs (Duvenaud et al., 2015; Liu et al., 2018; Segler et al., 2018; You et al., 2018; Jin et al., 2018; 2020a; Polykovskiy et al., 2020; Jin et al., 2020b; Satorras et al., 2021; Maziarz et al., 2021; Hoogeboom et al., 2022). There are two common ways of representing the structure of molecules: Simplified Molecular Input Line Entry System (SMILES) (Weininger, 1988) and molecular graphs (Bonchev, 1991). Graph neural networks can make effective use of the rich molecular graph representations by taking into account atoms, edges, and other structural information. SMILES strings convey less information about the molecular structure, but are more compatible with conventional sequence models (e.g., RNNs). Being able to generate valid molecules is the first step to AI-driven drug discovery and various solutions have been proposed to this problem. For example, GNN methods (Liu et al., 2018; Simonovsky and Komodakis, 2018; Jin et al., 2020a; Maziarz et al., 2021) can constrain the output space based on the chemical rules and SMILES-based approaches (Gómez-Bombarelli et al., 2018; Blaschke et al., 2018) can benefit from the abundant molecular data.

Besides structural validity, various chemical properties of the generated molecules, such as drug-likeness (QED) (Bickerton et al., 2012), Synthetic Accessibility (SA) (Ertl and Schuffenhauer, 2009) and molecular weight (MW) are critical factors when deciding whether candidate molecules can be synthesized in a laboratory and if they can be effective in real-world applications. A molecule's activity level on protein targets, whether to inhibit or to activate, is another very important property when treating specific diseases. A molecule that interacts successfully with the protein target is considered active and an activity score is measured based on how effective it is either to activate or inhibit the protein target's biological function. Researchers collected large molecule datasets, such as ChEMBL (Mendez et al., 2019) and ZINC (Irwin and Shoichet, 2005), that contain an array of bioactive molecules together with information about their properties and protein targets. By training on a curated set of molecules, the generative models can learn to generate new molecules that are similar in properties to those in the training set in order to produce novel drug candidates that satisfy multiple objectives, e.g. being drug-like and active against multiple protein targets. Benchmark metrics (Polykovskiy et al., 2020; Brown et al., 2019) are created to measure how similar the generated molecules are to the target dataset structurally and property-wise. The state-of-the-art results, however, show that there is still

room for improvements. Multi-objective generation by incorporating property predictors in the training pipeline (Jin et al., 2020b; Maziarz et al., 2021) is a promising avenue to address this type of problems, but there are also potential drawbacks. As Winter et al. (2019) summarized that, in drug discovery, the optimized objectives can be complex, conflicting, ill-defined or evolving over time. This could lead to improving some objectives while degrading others.

In this paper, we introduce an objective-agnostic and easy-to-implement technique to improve existing VAE-based molecule generation models – training additional stages of VAE's to generate latent representations for the previous-stage VAE. To show how this approach can enhance the manifold recovery of VAE models, we first study a simple MLP model trained on a synthetic sphere dataset. We then evaluate our method in an unconstrained molecule generation task and a fine-tuning task. In these experiments, we demonstrate the following claims:

- The multi-stage VAE is able to bring the properties of the generated molecules closer in distribution to the testing set in experiments on the ChEMBL dataset for unconstrained molecule generation;

- Fine-tuning the multi-stage VAE on the curated active molecules of two protein targets results in substantially more active outputs than fine-tuning only the first-stage model;

- In the tasks we described above, our method can achieve comparable or better results than specialized methods that directly optimize one or multiple target objectives using learned property predictors.

## 2 RELATED WORK

We structure our discussing based on the type of molecular representation underlying the individual methods. Most prior approaches fall into one of the following families – namely, the SMILES string approach, the molecular graph approach and the 3D point set approach. Many approaches have been proposed to generate molecules as SMILES strings (Segler et al., 2018; Gómez-Bombarelli et al., 2018). Kusner et al. (2017) and Dai et al. (2018) took advantage of the syntax of the SMILES strings and constrained the output of the VAE model in order to improve the validity of the generated molecules. Generative adversarial models have also been proposed to generate SMILES strings (Kadurin et al., 2017; Prykhodko et al., 2019; Guimaraes et al., 2017). Molecular graphs carry more information about the molecular structures than the SMILES string format, and GNN can effectively incorporate the additional information into the learning process (Duvenaud et al., 2015; Liu et al., 2018; Maziarz et al., 2021). Jin et al. (2018) proposed to generate molecular graphs in two steps – generate the tree-structured scaffolds first, and then combine these with the substructures to form molecules. Jin et al. (2020a) improved upon this prior result and proposed to generate new molecules via substructures in a coarse-to-fine manner to adapt to larger molecules, such as polymers. Satorras et al. (2021) introduced an equivariant graph neural network that can operate on molecular graphs. 3D representations of molecules are gaining traction in the research communities as they describe detailed spatial information of the molecules (Gebauer et al., 2019; 2022; Luo et al., 2021; Hoogeboom et al., 2022). However, none of the methods use the VAE framework. The goal of our paper is to improve upon the existing the VAE-based approaches. Other generative approaches to drug discovery include generative adversarial models (Kadurin et al., 2017; Prykhodko et al., 2019; Guimaraes et al., 2017) and diffusion models (Hoogeboom et al., 2022; Xu et al., 2022; Vignac et al., 2022).

## 3 METHOD

The VAE framework (Kingma and Welling, 2013) has enabled great success in the image generation domain and more recently VAE-based approaches have become popular for addressing the molecule generation problem. Many sophisticated architectures have been proposed to adapt it to molecular data (Kusner et al., 2017; Dai et al., 2018; Jin et al., 2019; Satorras et al., 2021). However, perfecting the underlying neural architecture does not remedy VAE's learning deficiency in manifold recovery (Dai and Wipf, 2019; Koehler et al., 2021). The *manifold hypothesis* (Fefferman et al., 2016) states that many high-dimensional real life data lie on low-dimensional manifolds embedded in high-dimensional ambient spaces. Koehler et al. (2021) found that the VAE is not guaranteed to recover the low-dimensional manifold where a nonlinear dataset lie. We show that a multi-stage VAE method can improve manifold recovery as demonstrated in a synthetic experiment (Figure 2) and further enhance the performance of pre-existing VAE models.

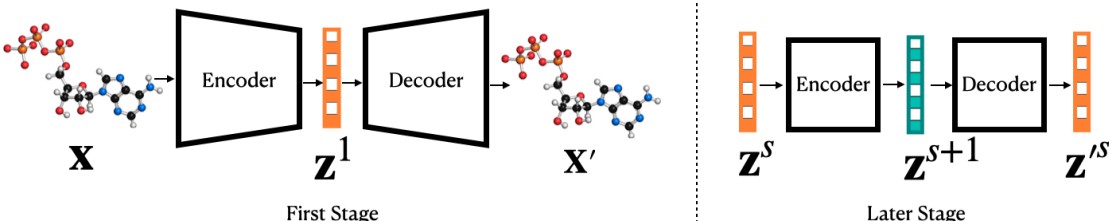

Figure 1: Overview of multi-stage VAE. In the first stage, the VAE trains on the molecule data $x_i$ and obtains the latent variables $z_i^1$ from $x_i$. The later-stage VAE is trained on the latent variables of the previous-stage VAE. $z_i^s$'s from the $s$-th stage VAE become the input to the $s+1$-th stage VAE during training. The later-stage VAE's input dimension is equal to the output dimension. During sampling, we sample $\mathbf{z} \sim \mathcal{N}(0, I)$ and obtain $z_i^s$ from the decoder. The output of a later-stage VAE decoder is used as the input for the previous-stage VAE decoder until the latent variable is decoded into a new molecule $x_i'$ in the first-stage.

## 3.1 VARIATIONAL AUTOENCODER

The variational inference framework assumes that the data $x$ is generated from a latent variable $\mathbf{z} \sim p(\mathbf{z})$. The prior $p(\mathbf{z})$ is assumed to be a multivariate standard normal distribution in the application of a VAE. Let $\phi$ be the variational parameters and $\theta$ denotes the generative parameters, the VAE model consists of a tractable encoder $q_\phi(\mathbf{z}|\mathbf{x})$ and a decoder $p_\theta(\mathbf{x}|\mathbf{z})$. A VAE model seeks to maximize the likelihood of the data, denoted as $\log p_\theta(\mathbf{x}) = \log \int p(\mathbf{z})p_\theta(\mathbf{x}|\mathbf{z})d\mathbf{z}$. However, the marginalization is intractable in practice due to the inherent complexity of the generator, or the decoder, thus an approximation of the objective is needed. The encoder and the decoder work together to approximate a lower bound to the log likelihood of the data. Ideally, by optimizing this lower bound we aim to increase the likelihood. This approximation enables the efficient posterior inference of the latent variable $\mathbf{z}$ given the input $x_i$ and for marginal inference of the output variable $\mathbf{x}$. The objective function of VAE consists of a KL divergence term $D_{KL}$ and a reconstruction term:

$$\mathcal{L}(\theta, \phi; \boldsymbol{x}) = -D_{KL}(q_\phi(\mathbf{z}|\mathbf{x}) \,\|\, p(\mathbf{z})) + \mathbb{E}_{q_\phi(\mathbf{z}|\mathbf{x})}[\log p_\theta(\mathbf{x}|\mathbf{z})] \leq \log p_\theta(\boldsymbol{x}) \tag{1}$$

For generation, latent representation $z_i$ is sampled from the prior $p(\mathbf{z})$ which is a multivariate standard normal and the decoder transforms $z_i$ into the output $x_i'$.

## 3.2 MULTI-STAGE VAE

Recovery of a low-dimensional data manifold embedded in a high-dimensional ambient space is a challenging task for VAE to perform. This provides an explanation for the discrepancy in properties between the generated molecules from existing VAE approaches and the testing dataset. Dai and Wipf (2019) hypothesized that training a continous VAE with a fixed decoder variance could add additional noise to the output. While training a VAE with a tunable decoder variance, they observed that the decoder variance has a tendency to approach zero and the diagonal entries of the encoder covariance to converge to either 0 or 1. A second VAE trained on the encoded latent variables of the first one with close to 0 decoder variance yields crisper and more realistic images than a one-stage VAE. By studying synthetically generated data with feedforward neural network, Koehler et al. (2021) showed that when the data is not linear, neither the manifold nor the density is guaranteed to be recovered by a one-stage VAE with tunable decoder variance. Without limiting ourselves to only 2 stages of VAE, we demonstrate in the following synthetic experiment that a multi-stage VAE can improve manifold recovery. We further apply the multi-stage VAE to the task of molecule generation involving much more complex architecture such as graph neural network and show that it is able to improve the distance between the property distribution of the test set and the generated set.

**Synthetic Experiment** We demonstrate that a multi-stage VAE setup improves the recovery of the manifold in a synthetic experiment with data generated from a ground-truth manifold (Koehler et al., 2021). We generated data from a 2-dimensional unit sphere such that the norms of all the generated data points are 1 (see Figure 2). The generated data is 3 dimensional but lies on the 2-dimensional unit sphere surface. The vectors are then padded with 16 dimensions of zeros to embed the data in an even higher-dimensional ambient space. The intrinsic dimension of the data is 2 and the ambient dimension is 19. We trained on this data in 3 stages – meaning, the encoded latent variables from the previous stage are used as input for training in the next stage. For the second and the third stage VAE, the latent dimension is set to be the same as their input (Figure 1 right). The decoder variance is tunable for all stages and the decoder variance

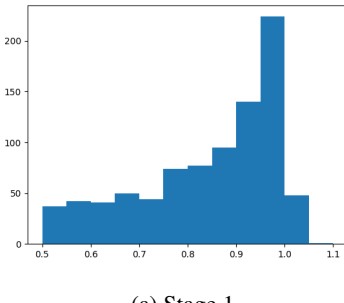

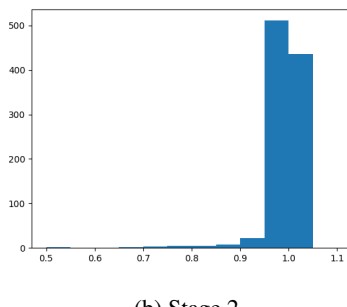

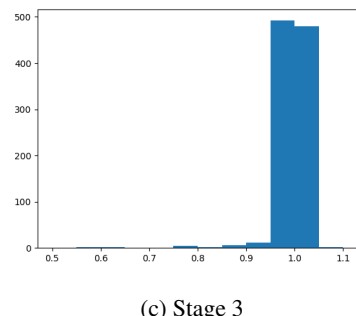

(a) Stage 1                                      (b) Stage 2                                  (c) Stage 3

Figure 2: Multi-stage VAE on synthetic data. The $x$-axis represents the norm of the data point and the $y$-axis represents the number of data points that are of $x$ distance away from the unit sphere center. The ground-truth data's histogram should be a Dirac delta at location 1. The figure for stage 1 shows that most of the generated points fall *inside* of the sphere instead. The sphere surface is mostly recovered and improved starting from stage 2.

of the first stage approaches 0 upon convergence. During sampling, the output of the later stage decoder is used as the input latent variable of the previous stage decoder. The last-stage VAE's latent variables are sampled from a standard normal distribution. We sample 1000 data points to visualize the results in the histograms. We observe that the VAE in the first stage *does not* recover the manifold and many of the generated data points fall *inside* of the sphere, echoing the finding by Koehler et al. (2021). In the second and third stage, we see that more data points fall *close to* the sphere, indicating a better recovery of the manifold. The third-stage has slightly fewer points falling below the distance of 0.95 to the origin than the second in this experiment.

**Application on Molecule Generation** The synthetic experiment is run on continuous data with a simple feed-forward architecture for both the decoder and the encoder. In this simple setting, two conditions on the first-stage VAE are satisfied for improvements in the later stage to occur (Dai and Wipf, 2019): i) the decoder variance converges to 0; and, ii) the entries in the diagonal of the encoder variance converge to either 0 or 1. The first condition can be applied to VAE's with multinomial setup common to molecule generation tasks and the decoder variance approaching zero is equivalent to the output being deterministic given the same latent variable. The second condition can also be verified because the latent space of the VAE models for molecule generation is generally continuous. However, the VAE architectures used for molecule generation are much more complex. Molecular data is discrete and the architectures for encoders and decoders can be hierarchical or sequential in their generation process. In Section 4, we examine how well each of the models we tested on fulfills the two conditions outlined for the continuous model. We also provide empirical studies on the suitability of a multi-stage VAE approach in the molecule generation domain by thorough evaluations on the samples' quality using structural and property statistics (Polykovskiy et al., 2020). We find that the multi-stage VAE helps to generate molecules that are more similar in property to the test set. Below we present the precise steps to train a multi-stage VAE (Figure 1):

1. Train a VAE on the molecular dataset $\{\boldsymbol{x}_i \,|\, i = 1, 2, \ldots n\}$, and upon convergence, save the latent variables $\mathbf{z}^1 \sim q_\phi(\mathbf{z}^1|\mathbf{x} = \boldsymbol{x}_i)$ for all the molecules in the dataset;

2. With $\{\boldsymbol{z}_i^1 \,|\, i = 1, 2, \ldots n\}$ as input, train additional stages of VAE with tunable decoder variance and the latent variable at stage $s$ is denoted as $\mathbf{z}^s$. We use feed-forward architectures for both the decoder $p_{\theta^s}(\mathbf{z}^s|\mathbf{z}^{s+1})$ and the encoder $q_{\phi^s}(\mathbf{z}^{s+1}|\mathbf{z}^s)$. They both follow Gaussian distributions. The dimension of $\mathbf{z}^s$ are the same across different stages. Repeat this step until the final stage, and the last-stage latent variable is denoted as $\mathbf{z}$.

3. During the sampling process, sample the latent representation of the last stage VAE, $\mathbf{z} \sim \mathcal{N}(0, I)$. Obtain the output from the stage-$s$ decoder $\mathbf{z}^s \sim p_{\theta^s}(\mathbf{z}^s|\mathbf{z}^{s+1} = \boldsymbol{z}_i'^{s+1})$ as the input to the stage-$(s-1)$ decoder. Repeat until we reach the first stage VAE and get the new molecule sample $\boldsymbol{x}_i'$ from the first stage decoder via $\mathbf{x} \sim p_\theta(\mathbf{x}|\mathbf{z}^1 = \boldsymbol{z}_i'^1)$.

We observe in our experiments that, for each additional stage of VAE we train, the decoder variance converges to a larger value than the previous one (details in Appendix B). And when the new stage's decoder variance converges to 1, the improvements become minimal. One way to interpret this method is that while the first-stage VAE learns a deterministic mapping between the latent representations and the molecular output, the later-stage VAEs learn to

map standard normal distribution to the distribution of the latent representation of the first-stage VAE. This is because the latent variables of the first-stage VAE do not necessarily follow the standard normal distribution as is generally assumed (Dai and Wipf, 2019). In the next section, we experiment on three pre-existing VAE models for general molecule generation task. We verify if they meet the two conditions outlined for the continuous synthetic setting earlier. And the models that do are able to generate molecules more similar to the test set by training additional stages of VAE. We also show that the multi-stage VAE can increase the number of active molecules generated when fine-tuned for a protein target.

## 4 EXPERIMENTS

In this section, we demonstrate the effectiveness of our methods on two generation tasks. First, our algorithm learns to generate molecules by training on a large molecular database (i.e., ChEMBL), and we show that training additional stages of VAE is able to generate molecules more similar to the test set in properties than training only a single stage. Our baselines include a model (Maziarz et al., 2021) that minimizes the losses between the true property values of the input molecules and the predicted property values of the output molecules for multiple targets during training and our method shows more consistent improvements across different properties. Second, our algorithm is fine-tuned on two curated molecular datasets that are active against two different protein targets (JAK2 and EGFR), and the multi-stage VAE improves the activity rate among the generated molecules. In this experiment, we compare our method against an RL algorithm (Jin et al., 2020b) using activity predictor as the reward function and show that we can achieve equivalent or better results in much shorter time.

### 4.1 UNCONSTRAINED GENERATION

We adopt three existing VAE model as our first-stage VAE – hierarchical GNN (Jin et al., 2020a), MoLeR (Maziarz et al., 2021), and character-level RNN (Polykovskiy et al., 2020) – to compare the effects of multi-stage VAE on different model architectures. We also adopt a GAN-based model (Prykhodko et al., 2019) as an additional comparison. We conduct experiments on the ChEMBL (Mendez et al., 2019) dataset – it consists of 1,799,433 bioactive molecules with drug-like properties. It is split into training, testing, validation datasets containing 1463k, 81k, and 81k molecules respectively. Details on our evaluation metrics are in Appendix A. Below we introduce the details of the models used in this study:

**Hierarchical GNN**: This method first extracts chemically valid motifs, or substructures, from the molecular graphs such that the union of these motifs covers the entire ChEMBL space. The model consists of a fine-to-coarse encoder that encodes from atoms to motifs and a coarse-to-fine decoder that selects motifs to create the molecules while deciding the attachment point between the motif and the emerging molecule. We use the configuration from the original model. The latent dimension of the VAE is 20 and we use 0.1 as the KL coefficient.

**MoLeR GNN**: Similarly to the hierarchical GNN, this method also extracts motifs in order to generate molecules piece by piece. The method's objective includes a regression term to match the true properties of the molecules and the predicted molecule properties from the latent variables. We implement our multi-stage method without such term and compare the property metrics of the generate molecules from our method to directly matching them in the training objective as "MoLeR + prop" in Table 1. We reduce the latent dimensions to 64 and use all the other original configuration.

**Vanilla RNN**: The inputs to the model are SMILES strings and the vocabulary consists of the low-level symbols in the SMILES strings. The encoder is a 1-layer GRU and the decoder is a 3-layer GRU. The latent dimension of the VAE is 128. We use the original configuration except a reduced KL coefficient of 0.01.

**Latent GAN**: This is also a 2-stage method. The first stage is a heteroencoder that takes SMILES strings as input while the second stage is a Wasserstein GAN with gradient penalty (WGAN-GP) that trains on the latent variables of the first stage encoder. The heteroencoder consists of an encoder and a decoder like an autoencoder and is trained with categorical cross-entropy loss. Afterwards, the GAN is trained to generate latent variables for the decoder of the heteroencoder. We use the original parameters for training.

We sample 10,000 molecules from each model to generate the results in Table 1. We include sample quality, structural and property statistics. The entries in the property statistics section are the *Wasserstein distances* between the property distribution of the test set and the generated set. A lower value in these statistics signals increased similarity to the test set in these properties. The results in the tables are averaged over 6 sets of samples generated with different random

| | Sample Quality | | | FCD ↓ | Structural Statistics | | | Property Statistics | | | |
|---|---|---|---|---|---|---|---|---|---|---|---|
| Stage # | Valid ↑ | Unique ↑ | Novelty ↑ | | SNN ↑ | Frag ↑ | Scaf ↑ | LogP ↓ | SA ↓ | QED ↓ | MW ↓ |
| HGNN#1 | 1.0 | 1.0 | 0.99 | 5.1 | 0.42 | 0.97 | 0.46 | $0.92_{0.016}$ | $0.070_{4.3e-3}$ | $0.024_{9.5e-4}$ | $68.8_{0.83}$ |
| HGNN#2 | 1.0 | 1.0 | 0.99 | 1.1 | 0.41 | 1.0 | 0.43 | $0.095_{0.019}$ | $\mathbf{0.069}_{5.8e-3}$ | $\mathbf{0.0067}_{1.0e-3}$ | $\mathbf{5.0}_{0.72}$ |
| HGNN#3 | 1.0 | 1.0 | 1.0 | 1.2 | 0.41 | 1.0 | 0.46 | $\mathbf{0.059}_{4.5e-3}$ | $0.069_{6.3e-3}$ | $0.016_{1.6e-3}$ | $7.7_{0.42}$ |
| MoLeR#1 | 1.0 | 1.0 | 0.99 | 2.1 | 0.41 | 0.96 | 0.48 | $0.16_{6.78e-3}$ | $\mathbf{0.028}_{1.96e-3}$ | $0.047_{2.78e-3}$ | $9.6_{0.70}$ |
| MoLeR #2 | 1.0 | 1.0 | 0.99 | 2.2 | 0.42 | 0.96 | 0.53 | $0.12_{6.13e-3}$ | $0.041_{4.31e-3}$ | $0.036_{9.25e-4}$ | $6.8_{0.71}$ |
| MoLeR #3 | 1.0 | 1.0 | 0.99 | 1.8 | 0.42 | 0.96 | 0.49 | $\mathbf{0.087}_{6.16e-03}$ | $0.031_{2.81e-03}$ | $\mathbf{0.028}_{2.31e-03}$ | $8.2_{4.0e-01}$ |
| MoLeR + prop | 1.0 | 1.0 | 0.99 | 2.1 | 0.43 | 0.97 | 0.49 | $0.11_{1.03e-02}$ | $0.13_{2.51e-03}$ | $0.033_{8.65e-04}$ | $\mathbf{6.6}_{4.80e-01}$ |
| RNN#1 | 0.86 | 1.0 | 1.0 | 1.84 | 0.38 | 1.0 | 0.38 | $\mathbf{0.088}_{7.8e-3}$ | $\mathbf{0.25}_{7.8e-3}$ | $\mathbf{0.0088}_{1.6e-3}$ | $3.2_{0.55}$ |
| RNN#2 | 0.87 | 1.0 | 1.0 | 1.86 | 0.38 | 1.0 | 0.36 | $0.099_{5.5e-3}$ | $0.27_{7.7e-3}$ | $0.0099_{1.5e-3}$ | $\mathbf{2.8}_{0.29}$ |
| LatentGan | 0.77 | 0.98 | 0.99 | 17.3 | 0.34 | 0.68 | 0.21 | $0.69_{0.019}$ | $0.63_{7.3e-3}$ | $0.047_{2.0e-3}$ | $27.2_{0.88}$ |

Table 1: Properties of the generated molecules trained on the ChEMBL dataset.

seeds from the model. We include the standard deviations for the property statistics but eliminated the rest as those are below 0.01. Stage #1 results are from the *original models*.

The HGNN#2 improves upon the first stage by many folds on property statistics. The most notable improvement from the ChEMBL dataset is the QED (from 0.024 to 0.0067), MW (from 68.8 to 5.0) and LogP (0.92 to 0.059). HGNN#3 does not show consistent improvement from HGNN#2. Structural statistics generally did not change a lot in the second and the third stage. The performance on these metrics of the later stages models may be bottle-necked by the first-stage graph decoder.

We trained three stages of the MoLeR model. The MoLeR#2 improves upon the MoLeR#1 in three (LogP, QED and MW) out of four metrics, while MoLeR#3 further improves upon MoLeR#2 in three (LogP, SA and QED) out of four metrics. Overall, MoLeR #3 improves upon MoLeR #1 in all property metrics except SA (from 0.028 to 0.031, only slightly higher). In LogP and QED, the metric goes down almost a half from 0.16 to 0.087 and from 0.047 to 0.028. The regression term in the "MoLeR + prop" optimizes over LogP, SA and MW. For two of these three metrics, "MoLeR + prop" reaches lower statistics than MoLeR#1, where such regression term is left out of the objective, but the SA metric of "MoLeR + prop" is *more than four times higher* than without the regression term. This highlights the challenges of directly matching multiple statistics at once during training – the objectives could be conflicting with each other and lead to unexpected results. Eventually, the MoLeR#3 reaches lower property statistics than "MoLeR + prop" in three (LogP, SA and QED) out of four metrics. For the only property metric (MW) our algorithm was worse (or bigger distance) in, it is only 24% worse than "MoLeR+prop" while for metric such as SA, our outputs are more than 300% better than "MoLeR+prop".

RNN#2 performs worse than RNN#1 in three out of four metrics. The RNN VAE's training process does not follow a standard VAE training procedure – the SMILES strings are input to both the encoder and decoder. This allows the decoder to rely less on the latent variables during the decoding process and the model exhibits signs of *posterior collapse* (Razavi et al., 2019; Fu et al., 2019) during training. Our hypothesis for the poor performance of the multi-stage VAE with the RNN model as the first-stage is that the variance of the first-stage decoder did not fulfill the condition of approaching 0 upon convergence and we will discuss more on that next.

**Verification on the Encoder and Decoder Variance Conditions** We investigate how well each of the three first-stage models fulfill the original conditions in a continuous setting for improvements in the later stages to occur. The examination can also help explain the different behaviors from them. We present our findings in Table 2.

1. *The decoder variance of the first-stage model converges to zero.* Variance in a discrete generation setting is not easy to measure, but we can substitute variance calculation with a simple experiment – input the same latent variable to the trained decoder for 1000 times, the number of distinct molecules the model generates can give us an indication if the decoder variance is approaching zero. We observe that for both of the GNN models, all 1000 identical latent variables generate 1000 identical molecules (Table 2 column "Decoder Diversity"). We can consider both models to have zero decoder variance. In contrast, the RNN model generates 1000 distinctive SMILES strings. Thus, its decoder variance did not approach 0 and our earlier hypothesis is verified.

2. *Each entry of the encoder variance diagonal either converges to 0 or 1.* We allow for 0.1 of tolerance around 0 and 1 and we find the following: The HGNN model's 20-dimensional encoder variance diagonal has all converged to 0; RNN's 128-dimensional encoder variance diagonal have converged to 111 of 1's, 15 of 0's;

and MoLeR's 64 dimensions of encoder variance diagonal have converged to 45 of 1's, 6 of 0's. The rest of the dimensions fall somewhere in between 0.1 and 0.9 as in Table 2's first three columns.

| Stage 1 Model | $x < 0.1$ | $0.1 \leq x \leq 0.9$ | $x > 0.9$ | Decoder Diversity |
|:---:|:---:|:---:|:---:|:---:|
| HGNN | 20 | 0 | 0 | 1 |
| MoLeR | 6 | 13 | 45 | 1 |
| RNN | 15 | 2 | 111 | 1000 |

Table 2: The first three columns document the number of encoder variance diagonal value $x$ that fall into either of the three categories: $x < 0.1$, $0.1 \leq x \leq 0.9$ and $x > 0.9$, and the last column document the number of unique SMILES by inputting 1000 identical latent vectors.

Overall, we see that the HGNN conforms to the conditions set for the simple continuous settings and the multi-stage VAE with HGNN as first-stage is able to improve significantly on the second stage and improve slightly in certain metrics on the third. MoLeR model is able to partially fulfill the conditions and the multi-stage VAE is still able to yield better results than the first-stage. The RNN model does not fulfill any of the two conditions and the outcomes from training the multi-stage VAE with RNN as the first-stage do not follow the synthetic experiment (Figure 2) that shows improvements in later stages.

## 4.2 GENERATION FOR A PROTEIN TARGET

In addition to unconstrained generation of molecules, we also explore generating molecules that target a specific protein. We pretrain the MoLeR model and two additional stages of VAE (with MoLeR as the first-stage) on the ChEMBL dataset and fine-tune them on two curated, and much smaller, datasets (Korshunova et al., 2022) consisting of molecules that are active inhibitors of Janus Kinas 2 (JAK2) and inhibitors of the Epidermal Growth Factor (EGFR). For each of the protein target, the regression dataset contains the molecules and their corresponding activity scores (from 0 to 10). A score above 6 is considered active. The dataset size is around 19k for JAK2, in which around 15.6k are active, and around 15k for EGFR, in which around 7.8k are active. The full dataset is used to train a regressor of the activity score while only the active molecules are used to fine-tune the VAE. We divide each dataset by 80%, 10% and 10% for training, validation and testing purposes for both the VAE and the predictor. In addition, there is a separate classification dataset for each protein target – 60k for JAK2 and 50k for EGFR – containing only the binary activity information of each molecule. 10% of the dataset is used for testing the classifiers.

**Metrics** We evaluate our method and baseline methods in three general categories: activity, diversity and novelty (Table 3 and 4). All evaluations are reported with means and standard deviations across 5 random seeds used to generate 5 datasets with 1000 molecules each. Particularly for activity scores, we report both the mean score and percentage of active molecules (6 as the cutoff point) in a dataset from two different regressors – a Chemprop (CPR) (Yang et al., 2019) model, and a Random Forest (RFR) model with Morgan fingerprint features (Rogers and Hahn, 2010). Both regressors reach an RMSE of 0.5 for the JAK2 dataset and 0.6 for the EGFR dataset. In addition to the regressors, we also train two classifiers using Chemprop (CPC) and Random Forest (RFC) with Morgan fingerprint features on the classification dataset and we report the percentage of active molecules in the generated set by the classifiers as well. All predictors' performances on the active test set split are included in Table 3 and 4 as a reference. Novelty is defined as the fraction of molecules with its nearest neighbor similarity in the active training set lower than 0.4. Diversity is calculated based on the pairwise molecular distance $\text{sim}(X,Y)$ within the generated datset. The function $\text{sim}(\cdot, \cdot)$ is defined as the Tanimoto distance over Morgan fingerprints of two molecules. We define the two metrics as follows (Jin et al., 2020b):

$$\text{Diversity} = 1 - \frac{2}{n(n-1)} \sum_{X,Y} \text{sim}(X,Y) \quad \text{Novelty} = \frac{1}{n} \sum_{\mathcal{G}} \mathbf{1}[\text{sim}(\mathcal{G}, \mathcal{G}_{\mathcal{SNN}}) < \mathbf{0.4}] \qquad (2)$$

**Methods** We compare the generated molecules from the fine-tuned multi-stage MoLeR model to the fine-tuned one-stage MoLeR model. We include RationaleRL (Jin et al., 2020b) as a reinforcement learning-based baseline. We obtained the multi-stage MoLeR from training on the full ChEMBL dataset as described in Section 3.2. The first-stage (also the original) MoLeR model was fine-tuned on the curated active molecules dataset after pretraining on the full ChEMBL dataset. The second and third-stage VAEs are initialized with the parameters from the corresponding

stage of the pretrained multi-stage MoLeR and fine-tuned on the encoded latent variables of the curated dataset from the fine-tuned previous-stage VAE. We fine-tuned the later-stage VAE's in three ways: fine-tuning the entire model, fine-tuning only the two inner-layers connecting to the latent sampling layer, and fine-tuning only the two outer-layers connecting to the input and output. We visually compare the distributions of the activity scores for EGFR as predicted by the Chemprop model from the three types of fine-tuning methods and baselines in Figure 3. The quantitative metrics are shown in Table 3 and 4. Each evaluation reports metrics for the active test set as a reference.

| Model Type | Activity (CPC) | Activity (CPR) | Mean (CPR) | Activity (RFC) | Activity (RFR) | Mean (RFR) | Diversity | Novelty(0.4) | Time [1] |
|---|---|---|---|---|---|---|---|---|---|
| Active Test set | 0.93 | 0.92 | 7.4 | 0.98 | 0.89 | 7.1 | 0.85 | 0.016 | - |
| MoLeR | $0.44_{1.18e-02}$ | $0.57_{1.21e-02}$ | $6.3_{2.94e-02}$ | $0.41_{1.14e-02}$ | $0.60_{1.46e-02}$ | $6.3_{2.16e-02}$ | $0.87_{2.26e-03}$ | $0.63_{1.81e-02}$ | 4 min |
| RationaleRL | $0.79_{7.94e-03}$ | $0.84_{1.45e-02}$ | $6.8_{1.56e-02}$ | $0.88_{1.23e-02}$ | $0.94_{4.50e-03}$ | $6.7_{1.42e-02}$ | $0.79_{1.38e-03}$ | $0.055_{5.24e-03}$ | 4.5 d |
| Whole-Model | $0.53_{6.43e-03}$ | $0.63_{6.65e-03}$ | $6.4_{3.18e-02}$ | $0.50_{1.38e-02}$ | $0.69_{4.31e-03}$ | $6.5_{1.91e-02}$ | $0.85_{9.95e-04}$ | $0.54_{9.65e-03}$ | 1.5 h |
| Inner-Layer | $0.79_{1.61e-02}$ | $0.87_{1.31e-02}$ | $7.4_{3.92e-02}$ | $0.88_{1.27e-02}$ | $0.90_{8.40e-03}$ | $7.0_{2.04e-02}$ | $0.71_{4.41e-03}$ | $0.17_{2.03e-02}$ | 1.25 h |
| Outer-Layer | $0.85_{1.30e-02}$ | $0.89_{5.31e-03}$ | $7.2_{3.17e-02}$ | $0.86_{1.75e-02}$ | $0.89_{1.17e-02}$ | $6.9_{2.61e-02}$ | $0.71_{5.30e-03}$ | $0.22_{8.78e-03}$ | 1.5 h |

Table 3: Evaluation of the generated molecules targeting EGFR by three multi-stage VAE fine-tuning methods: fine-tuning the whole model, fine-tuning only the inner-layers and fine-tuning the outer-layers. They are compared against baseline models such as fine-tuned one-stage MoLeR and RationaleRL. The evaluation metrics include percentage of active molecules, mean activity scores, diversity and novelty.

| Model Type | Activity (CPC) | Activity (CPR) | Mean (CPR) | Activity (RFC) | Activity (RFR) | Mean (RFR) | Diversity | Novelty(0.4) | Time [1] |
|---|---|---|---|---|---|---|---|---|---|
| Active Test set | 0.97 | 0.97 | 7.5 | 0.99 | 0.98 | 7.4 | 0.88 | 0.016 | - |
| MoLeR | $0.52_{1.15e-02}$ | $0.74_{6.89e-03}$ | $6.5_{1.17e-02}$ | $0.44_{1.73e-02}$ | $0.89_{1.24e-02}$ | $6.7_{2.44e-02}$ | $0.90_{5.62e-04}$ | $0.85_{6.71e-03}$ | 7 min |
| RationaleRL | $0.85_{8.96e-03}$ | $0.79_{1.67e-02}$ | $6.6_{2.93e-02}$ | $0.92_{4.93e-03}$ | $0.96_{7.26e-03}$ | $6.8_{1.67e-02}$ | $0.87_{2.58e-03}$ | $0.24_{7.42e-03}$ | 8.5 d |
| Whole-Model | $0.54_{8.31e-03}$ | $0.78_{1.02e-02}$ | $6.6_{2.22e-02}$ | $0.46_{1.13e-02}$ | $0.91_{5.90e-03}$ | $6.7_{1.16e-02}$ | $0.89_{4.47e-04}$ | $0.82_{7.80e-03}$ | 2.8 h |
| Inner-Layer | $0.66_{7.58e-03}$ | $0.88_{1.11e-02}$ | $6.9_{1.01e-02}$ | $0.56_{6.68e-03}$ | $0.94_{2.28e-03}$ | $6.8_{1.36e-02}$ | $0.88_{8.39e-04}$ | $0.78_{6.29e-03}$ | 2.8 h |
| Outer-Layer | $0.62_{2.27e-02}$ | $0.89_{8.37e-03}$ | $6.9_{3.72e-02}$ | $0.65_{1.23e-02}$ | $0.95_{1.96e-03}$ | $6.9_{1.53e-02}$ | $0.86_{1.29e-03}$ | $0.69_{5.37e-03}$ | 2.8 h |

Table 4: Evaluation of the generated molecules targeting JAK2 by by three multi-stage VAE fine-tuning methods: fine-tuning the whole model, fine-tuning only the inner-layers and fine-tuning the outer-layers. They are compared against baseline models such as fine-tuned one-stage MoLeR and RationaleRL. The evaluation metrics include percentage of active molecules, mean activity scores, diversity and novelty.

For both of the protein targets, fine-tuning the multi-stage VAE in either of the three ways produces more active molecules of the protein targets than fine-tuning only the first-stage VAE (Table 3 and 4) with all predictors considered. The improvement is especially pronounced on the EGFR dataset, which is about half of the JAK2 dataset size. Particularly, fine-tuning only the inner-layers or outer-layers produces more active molecules than fine-tuning the whole model. Their activity score distributions are also more similar to the test set's (Figure 3). The peak of the activity score distribution around 7 is well-captured by the two fine-tuning methods of the VAE in Figure 3a and 3b. Fine-tuning only two layers reduces the number of parameters to be trained with the same amount of data, thus leads to better results.

RationaleRL model is initialized with a generative model trained on the ChEMBL dataset and then fine-tuned with an RL algorithm. The Random-Forest classifier with Morgan fingerprint features is used as its reward signal (Jin et al., 2020b) in the fine-tuning process. RationaleRL generally has higher percentage of active molecules when predicted by the two Random Forest models than Chemprop, a sign of possible overfitting. No such pattern is observed for our methods that did not involve property predictors in the fine-tuning process. On the EGFR dataset, RationaleRL reaches higher level of activity than fine-tuned multi-stage VAE as predicted by Random Forest models but lower level of activity when predicted by Chemprop. On the JAK2 dataset, our methods have lower activity level than RationaleRL as predicted by both classifiers but both regressors have predicted similar or slightly higher activity levels on the generated molecules from our methods than RationaleRL. Fine-tuned multi-stage VAE is similar to RationaleRL on diversity but higher on novelty metric. Our method reaches slightly higher mean activity scores than RationaleRL on both datasets predicted by both regressors. In addition, RationaleRL is much more computationally intensive to train.

Multiple factors could contribute to the large discrepancy between regressors and classifiers' predictions on the dataset generated by our method for JAK2. It could be that the classifiers are more adept at identifying negative samples due to the larger training set, which also includes more negative examples. Another reason could be that the predictors fail to generalize to an out-of-distribution dataset, such as the generated one, despite performing well on the ground-truth data. This is evidenced by the high novelty scores from the generated sets of fine-tuned MoLeR and its multi-stage versions in Table 4. These differing results highlight the ambiguity in drawing conclusions based on property predictors, which further complicates the optimization process.

---

[1]Training on a single "NVIDIA GeForce RTX 2080 Ti" GPU.

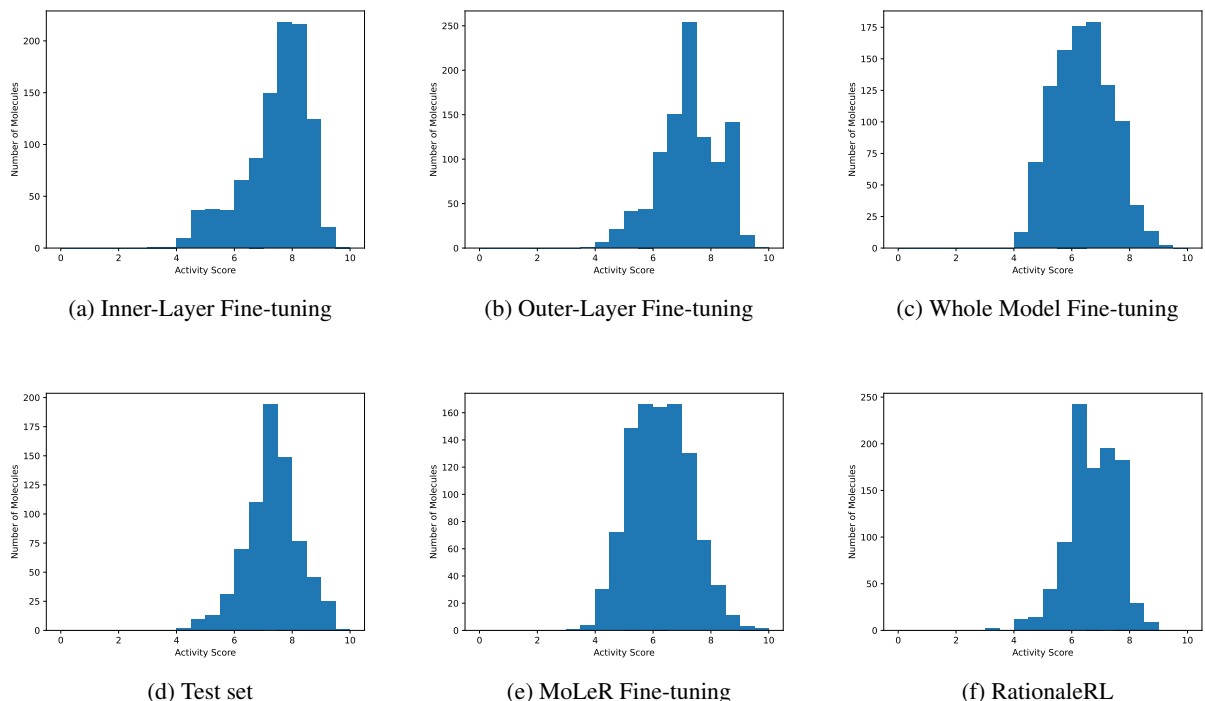

Figure 3: The distributions of the generated molecules' activity scores on the EGFR protein predicted by the Chemprop model. We include 5 sets of molecules generated by different methods as well as the ground-truth test set. Additional activity score histograms can be found in Appendix D

## 5 DISCUSSION

In this paper, we present a novel multi-stage VAE model that enhances the properties of the generated molecules from pre-existing VAE models. It achieves the goal in an objective-agnostic way, meaning, our method does not optimize over any particular property objectives during training. Yet, it is able to achieve comparable, and sometimes better outcomes than specialized approaches that directly optimizes over one or multiple property objectives with the help of property predictors. We demonstrate this in two experiments: 1) an unconstrained generation experiment trained on ChEMBL dataset in which we compare against MoLeR (Maziarz et al., 2021) that includes regression terms in the VAE objective to match the true property values of the input molecules with the predicted propertiy values of the output molecules; and 2) a generation for protein target experiment where our multi-stage VAE is fine-tuned on two active molecule datasets against two protein targets. We compare our method to RationaleRL (Jin et al., 2020b), which uses an activity predictor as the reward function. Our findings also bring up the complications in using property predictors for multi-objective generation, e.g. overfitting, conflicting objectives, and differing prediction results from different predictors. Addressing these issues is an important direction for future research.

