# OpenReview forum: "Objective-Agnostic Enhancement of Molecule Properties via Multi-Stage VAE"
_ICLR.cc/2024/Conference — ICLR 2024 Conference Withdrawn Submission_

### Official Review · Reviewer_DTvh · 2023-10-27

**Soundness:** 1 poor
**Presentation:** 1 poor
**Contribution:** 1 poor
**Rating:** 1
**Confidence:** 5

**Summary:**

The authors propose a new method for training VAEs for drug discovery tasks. Unfortunately, I must vote to reject without further review as the authors blatantly violated the ICLR submission template. Violations include:

* Obviously significantly reducing the margin size to fit within the page limit.
* Removing the "Under review as a conference paper at ICLR 2024" header.
* Not including a references section in their submission.

**Strengths:**

Rejecting without further review.

**Weaknesses:**

Rejecting without further review.

**Questions:**

Rejecting without further review.

---

### Official Review · Reviewer_uwWu · 2023-10-30

**Soundness:** 1 poor
**Presentation:** 2 fair
**Contribution:** 1 poor
**Rating:** 3
**Confidence:** 4

**Summary:**

The article "Objective-agnostic enhancement..." describes the use of multi-layer VAE models for the prediction and generation of small molecules for drug discovery problems. The use of multi-layer VAE models is proposed in view of their better generative statistics, approximation of the true, for a given dataset, immersion in high-dimensional space.

The article is very simple, offering basically nothing new. The level of workmanship is low, additionally parts of the paper are clearly missing.

I believe that the proposed article is not ready, does not contain any novelties, and as such should not be accepted.

**Strengths:**

1. Correct posing of a problem, with correct references to related work.
2. The authors undertake the issue of finding better low-dimension space immersion in high-dimensional space. On the other hand, it is not proved that their solution is the right one.

**Weaknesses:**

1. The authors note that typically, high-dimensional data can be projected onto a low-dimensional space immersed in a high-dimensional space. To demonstrate this, they conduct a very simple experiment on synthetic data. This experiment does not show much, only the authors claim that for one of the later layers a space with the correct dimensionality for the data is obtained. This experiment is clearly spurious and can be removed.
2. The authors propose to build a VAE model with many hidden layers iteratively. In each step, developing the projection of the most recently obtained hidden layer into a new VAE model. The obtained model would get the variances on the diagonal reach the values {0, 1}, which would show the selection/reduction of individual dimensions projection.
It seems that the analogy with synthetic experiment is insufficient for a direct application of this approach.
3. Several models with randomly selected numbers of hidden layers are used in the experiments.
4. In the comparisons, each model has a different dimensionality of hidden layers for the same data.
5. There is no bibliography in the work. I suspect that this is a simple editorial error (references in the work are marked as links), but such an error in a work for a major conference is unacceptable. Similarly, there is no appendix to which there is a reference in the main text.

**Questions:**

1. Is the synthetic experiment really needed? I would remove it if I were you. In place of it, I would give some better mathematical discussion.
2. Is the analogy between the scheme presented in the synthetic data experiment to the problem of molecule generation really valid? Could you, please, better justify the analogy?
3. Why just that number of layers are used in the models? Why are there 3 layers for HGNN and MoLeR? The "MoLeR + prop" model is completely unclear? Why does the RNN model only have two layers? The LatentGAN model seems to have only one hidden layer, right? Or maybe some of the nodels are for comparison? This is not clear from the text of the work.
4. If one of the goals was to show that multi-layer models, along with variance optimization, allow for better matching of the implicit dimensions, does using different dimensions allow this to be shown for different models? There is no description of the individual statistics used in the work.
5. Was variance optimization used for all models? This is not clear from the text.
6. The standard in the editorial composition of scientific works is to place descriptions of tables above them, not below.

---

### Official Review · Reviewer_un1d · 2023-11-02

**Soundness:** 2 fair
**Presentation:** 2 fair
**Contribution:** 2 fair
**Rating:** 1
**Confidence:** 4

**Summary:**

A multi-stage VAE model that help generate molecules with conditional properties.

**Strengths:**

Proposed a novel multi-stage model.

**Weaknesses:**

No reference section.
No under review header.
I believe this paper is not ready to be reviewed.

**Questions:**

N/A